# DivKnowQA: Assessing the Reasoning Ability of LLMs via Open-Domain Question Answering over Knowledge Base and Text

## Abstract

Large Language Models (LLMs) have exhibited impressive generation capabilities, but they suffer from hallucinations when solely relying on their internal knowledge, especially when answering questions that require less commonly known information. Retrieval-augmented LLMs have emerged as a potential solution to ground LLMs in external knowledge. Nonetheless, recent approaches have primarily emphasized retrieval from unstructured text corpora, owing to its seamless integration into prompts. When using structured data such as knowledge graphs, most methods simplify it into natural text, neglecting the underlying structures. Moreover, a significant gap in the current landscape is the absence of a realistic benchmark for evaluating the effectiveness of grounding LLMs on heterogeneous knowledge sources (e.g., knowledge base and text). To fill this gap, we have curated a comprehensive dataset that poses two unique challenges: (1) **Two-hop multi-source questions** that require retrieving information from both open-domain structured and unstructured knowledge sources; retrieving information from structured knowledge sources is a critical component in correctly answering the questions. (2) **Generation of symbolic queries** (e.g., SPARQL for Wikidata) is a key requirement, which adds another layer of challenge. Our dataset is created using a combination of automatic generation through predefined reasoning chains and human annotation. We also introduce a novel approach that leverages multiple retrieval tools, including text passage retrieval and symbolic language-assisted retrieval. Our model outperforms previous approaches by a significant margin, demonstrating its effectiveness in addressing the above-mentioned reasoning challenges.

## 1 Introduction

LLMs have shown exceptional performance in multi-hop question-answering (QA) tasks over text (TextQA) (Rajpurkar et al., 2018; Kwiatkowski et al., 2019; Joshi et al., 2017; Trivedi et al., 2022a; Yang et al., 2018; Ho et al., 2020), tables (TableQA) (Yu et al., 2018; Zhong et al., 2017; Pasupat and Liang, 2015; Chen et al., 2019), and knowledge-bases (KBQA) (Gu et al., 2021; Yih et al., 2015; Talmor and Berant, 2018; Bao et al., 2016), where the supporting fact is contained in a single knowledge source – structured or unstructured. However, in many real-world scenarios, a QA system may need to retrieve information from both unstructured and structured knowledge sources; failing to do so results in insufficient information to address user queries.

While existing QA benchmarks provide diverse perspectives for evaluating models (Table 1), they are limited in assessing the performance of retrieval-augmented language models across heterogeneous knowledge sources in the following aspects: *(1) Closed-book QA:* Closed-book questions do not accurately reflect the real-world setting where individuals generally have access to diverse knowledge sources on the Internet; *(2) Automatically generated data:* The lack of human verification results in erroneous data; *(3) Imbalanced emphasis across different knowledge sources:* Current benchmarks feature knowledge sources with varying levels of importance. Answers may be found in multiple sources, leading models to prioritize textual sources while underutilizing structured knowledge sources; *(4) Suboptimal use of structured knowledge:* Structured knowledge sources are typically treated as textual sources by linearizing triplets from the knowledge base or rows/columns from

tables, missing the opportunity to fully realize the benefit of highly-precise structured knowledge by probing them via symbolic queries.

Despite the inherent challenges, being able to generate structured queries effectively can offer a number of benefits. First, unlike a query to retrieve text passages, the structured query itself can share the responsibility of reasoning Liu et al. (2022). For example, for the question "`How many awards has Neil Armstrong received?`", to get an answer from a knowledge base such as Wikidata Vrandečić and Krötzsch (2014), a SPARQL query Pérez et al. (2006) can use an aggregation function to return the numerical number as the final result as shown in Appendix A.5. In contrast, a text retriever needs to locate all the relevant passages and rely on a reader module to get the final result. The commonly used readers often come with an input length constraint. The number of returned passages could be too many to fit into the reader's context, causing a wrong answer. Even when the context length is not an issue, even the best LLMs have difficulties in locating the answer Liu et al. (2023a). Besides, there is less room for ambiguity in structured queries. For example, a dense retriever cannot easily distinguish between similar song titles such as "`I'll be good to you`" and "`I have been good to you`" by different singers. On the other hand, given the right identifier of the entity, the structured knowledge search engine can return the relevant information for the exact entity.

In this work, we propose DIVKNOWQA, a novel fact-centric multi-hop QA benchmark that requires models to utilize heterogeneous knowledge sources equitably in order to answer a question. We perform the first study to assess the reasoning ability of LLMs, via jointly exploiting open-domain QA over heterogeneous knowledge sources. In particular, we have chosen Knowledge Base (KB) as our primary case study for the structured source, and we have created a dataset comprising 940 human-annotated examples. Additionally, each entry in our dataset includes a corresponding symbolic SPARQL query to facilitate the retrieval of information from the KB. To generate the questions, we construct a question collection pipeline comprising three key steps: text-based QA sampling, KB question generation, and question composition, all while minimizing the need for human annotation efforts.

To set up a baseline, in addition to benchmarking on standard and tool-augmented LLMs, we propose a Diverse rEtrieval Tool Augmented LLM (DETLLM) to address the challenges posed by DIVKNOWQA. DETLLM decomposes a multi-hop question into multiple single-hop questions, and adopts two novel strategies: (1) *symbolic query generation* to retrieve supportive text from a KB by transforming a single-hop natural question into a SPARQL query, and (2) *retrieval tool design*, which includes a textual retriever and a symbolic query generation tool to recall relevant evidence from heterogeneous knowledge sources. Our method shows improvements of up to 4.2% when compared to existing methods.

## 2 THE DIVKNOWQA DATASET

### 2.1 DATASET COLLECTION

Our goal is to create a method for generating complex questions from diverse knowledge sources, making each source indispensable; and we aim to do so with a minimal human annotation effort. Additionally, we wish to provide Wikidata entity and relation IDs to support structured query-based knowledge retrieval. Figure 1 depicts our proposed method. We first sample a single-hop text question from the Natural Question dataset (Kwiatkowski et al., 2019) as an anchor, to which we link to a relevant Wikidata triplet. Then single-hop KB questions are generated based on the sampled triplets thereby using the anchor question to automatically compose a *heterogeneous* multi-hop question. Human annotators finally verify the quality of the machine-generated question and rewrite the question that needs revision. In the following, we elaborate on the steps.

**Natural Questions as Anchors** The Natural Question (NQ) dataset is a question-answering dataset containing tuples of (`question, answer, title, passage`), where `title` and `passage` are respectively the title of the Wikipedia page and the passage containing the answer. The questions in NQ were collected from real-world user queries issued to the Google search engine, and it contains 307K training examples. We concentrate on constructing a multi-hop dataset linked through the initial step's single-hop answer. To achieve this, we extract question-answer pairs where the

Table 1: Comparing benchmarks for heterogeneous question-answering tasks. The column OpenR stands for open information retrieval, Human for human-written questions, EI for equitable importance of knowledge sources, and SGT for structured ground truth.

| Dataset | KB | Text | Table | OpenR | Human | EI | SGT |
|---|---|---|---|---|---|---|---|
| HybridQA (Chen et al., 2020) | ✗ | ✓ | ✓ | ✗ | ✓ | ✓ | ✗ |
| OTT-QA (Chen et al., 2021a) | ✗ | ✓ | ✓ | ✓ | ✓ | ✓ | ✗ |
| NQ-Tables (Herzig et al., 2021) | ✗ | ✓ | ✓ | ✗ | ✗ | ✗ | ✗ |
| TAT-QA (Zhu et al., 2021) | ✗ | ✓ | ✓ | ✗ | ✓ | ✓ | ✗ |
| MultimodelQA (Talmor et al., 2021) | ✗ | ✓ | ✓ | ✗ | ✓ | ✗ | ✗ |
| Manymodelqa (Hannan et al., 2020) | ✗ | ✓ | ✓ | ✗ | ✓ | ✗ | ✗ |
| FinQA (Chen et al., 2021b) | ✗ | ✓ | ✓ | ✗ | ✓ | ✓ | ✗ |
| HetpQA (Shen et al., 2022) | ✗ | ✓ | ✓ | ✗ | ✓ | ✗ | ✗ |
| CompMix (Christmann et al., 2023) | ✗ | ✓ | ✓ | ✓ | ✓ | ✗ | ✗ |
| WikiMovies-10K (Miller et al., 2016) | ✓ | ✓ | ✗ | ✓ | ✓ | ✗ | ✗ |
| MetaQA (Zhang et al., 2018) | ✓ | ✓ | ✗ | ✓ | ✓ | ✗ | ✗ |
| DIVKNOWQA (Ours) | ✓ | ✓ | ✗ | ✓ | ✓ | ✓ | ✓ |

question contains a succinct answer of up to $5$ words to ensure the quality of the resulting composed question.

**Linking Natural Questions to Wikidata**  We adopt the notion of *bridge entity* from Yang et al. (2018) to describe the single-hop answer in the initial step when breaking down a multi-hop question. We explore two linking options, each involving a unique choice of bridging an entity to connect the natural question to Wikidata. We explain the options using the example question "`Who plays Mary Poppins in Mary Poppins Returns?`" with the answer "`Emily Blunt`". (a) Text → KB Approach: We treat the answer "`Emily Blunt`" as the bridge entity, and search for a Wikidata triplet where "`Emily Blunt`" is the *subject*, for example, (`Emily Blunt, sibling, Felicity Blunt`). (b) KB → Text Approach: In this alternative method, we recognize the question entity that exists in Wikidata, in this case, "`Mary Poppins Returns`", as the bridge entity. For simplicity, we only consider the entity mentioned in the Wikipedia title. We then link to the Wikidata triplet using it as the *object*, leading to triplets such as "(`William Weatherall Wilkins, present in work, Mary Poppins Returns`)".

**Selection of KB Triplets**  To maintain an equal emphasis on both structured and unstructured knowledge sources, we implement a meticulous selection process for KB triplets to ensure that the associated knowledge cannot be easily obtained by merely retrieving information from the textual source (Wikipedia passages). We retain triplets $(sub, relation, obj)$, where either the subject $sub$ is not linked to a Wikipedia page or the object $obj$ does not exist within the Wikipedia page associated with the subject. This ensures that simply retrieving the Wikipedia passage for the $sub$ is unlikely to yield an answer to a question involving $sub$ and $obj$, thereby requiring the model to utilize the KB. Furthermore, when generating questions in the KB → Text linking option, we selectively retain triplets where only one object is associated with the given relation and subject This approach ensures the completeness and uniqueness of the reasoning chain. For example, given a composed question, "Who plays Mary Poppins in Lin-Manuel Miranda's notable work?", "Mary Poppins Returns" is one of the notable works from "Lin-Manuel Miranda". By querying KB given the subject "Mary Poppins Returns" and the relation "notable work", we will locate multiple answers rather than the single bridge entity "Mary Poppins Returns", posing a challenge to infer the second sing-hop question "Who plays Mary Poppins in Mary Poppins Returns?".

**Generating Single-Hop KB Questions**  We then create single-hop questions from the selected $(sub, relation, obj)$ triplets. These questions are designed to emphasize the relationship between $sub$ and $obj$, with $obj$ being the expected answer. For instance, for the KB triplet "(`Emily Blunt, sibling, Felicity Blunt`)", we expect to generate a question like "`Who is the sibling of Emily Blunt?`". For this, we employ the `gpt-turbo-3.5` LLM from OpenAI; the prompt can be found in Appendix A.1.

**Generating Heterogeneous Multi-Hop Questions**  In this stage, we wish to create a multi-hop question by composing a textual question and a KB question. We generate such *heterogeneous* questions by carefully chaining two single-hop questions together. DIVKNOWQA supports three

question types: short entity, yes/no, and aggregate questions, and two question composition orders: Text → KB and KB → Text. This combination results in a total of five question types, as we construct aggregate questions following only the Text → KB order. We employ `gpt-turbo-3.5` as a question composer to connect two single-hop questions. This is achieved by substituting the entity mentioned in the outer question with a rephrased version of the first question. The prompt for generating the multi-hop questions is given in Appendix A.2. Our generation method for different question types is discussed as follows.

**Short Entity Question**   We use a factoid entity as the final answer. The final answer can be the object from Wikidata or the factoid answer from NQ.

**Yes/No Question**   In contrast to Short Entity questions, Yes/No questions involve an additional step. Initially, the original question is reformulated into a verification-style question typically starting with phrases like "`Is/Was/Were/Does/Do/Did`". This new question includes a candidate answer for verification purposes. For instance, let's consider the original question "`What grade were they in High School Musical 1?`" with a known answer of "`juniors`". To create a verification question, we might rephrase it as "`Were they seniors in High School Musical 1?`" and include the verifying answer "`seniors`" within the question. Generating the candidate answer for verification can be a complex task as it requires choosing a verifying answer that aligns well with the context of the question. Sampling incorrect distractors as verifying answers is also a part of the process. These distractors should be incorrect but closely related to the answer, and they are generated by prompting `gpt-turbo-3.5`. This approach ensures that the verification process is robust and accurate, preventing situations where the verifying answer deviates from the question's context and potentially leads to a simplistic answer "`no`" during evaluation.

**Aggregate Question**   We formulate aggregating questions in the "`Text → KB`" composition order, where the outermost question pertains to counting the number of associated triplets based on the given subject and relation. For instance, the outermost question "`How many awards does Milton Friedman receive?`" arises from the KB triplets of the form `(Milton Friedman, award received, award name)`" with 10 such `award name` objects. In such cases, we leverage the aggregate feature offered by the structural query (i.e., SPARQL).

## 2.2   HUMAN ANNOTATION

We recruited five individuals, three undergraduate and two graduate students with experience in the field of NLP for data verification and annotation. Each question underwent a verification and rewriting process involving two annotators. To mitigate any potential annotation bias, we presented each question to both annotators, with the order of examples shown to annotators randomized. Annotators were tasked with assessing the quality of KB question generation, and they had three options to choose from: "Accept", "Revise", or "Reject". When a question required revision, annotators were instructed to make modifications while preserving the focus on the subject and relation and keeping the answer unchanged. Additionally, they were responsible for evaluating the quality of complex questions and providing necessary revisions. The instruction provided to human annotators is shown in Appendix A.4. Annotators were duly compensated for their valuable contributions to our study. Out of 1,000 examples that were annotated, 757 examples received unanimous approval, 183 underwent revisions, and 60 were rejected. Both unanimously accepted and revised examples were included in the dataset.

## 2.3   DATASET STATISTICS AND ANALYSIS

In this section, we analyze the question types and KB single-hop relation types in DIVKNOWQA.

**Question Central Word**   Taking inspiration from (Yang et al., 2018), we designate the first three words of a question as the Center Question Words (CQW. We adopt this approach because our questions typically do not contain comparison queries, and a majority of question words are found at the beginning of the question. Appendix A.6(a) provides a visual representation of CQW in DIVKNOWQA.

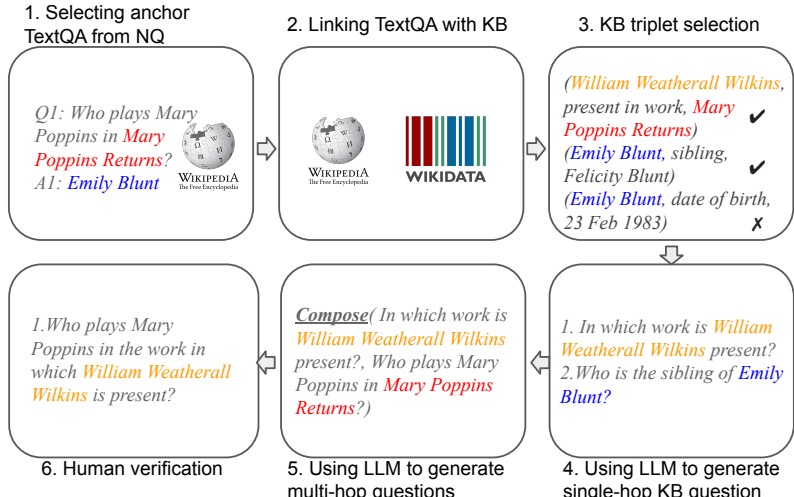

Figure 1: An overview of DivKnowQA data generation process.

**KB Relation Types** We also analyze the distribution of relations by counting the frequency of different relations that appear in the KB triplets used to construct the single-hop KB questions. Appendix A.6(b) features the distribution of diverse relations.

**Anecdotal Examples for Representative Types** In A.7, we present illustrative examples drawn from the DivKnowQA benchmark for each of our five question composition types. These examples serve to showcase how our dataset necessitates information retrieval from diverse sources in varying orders. Additionally, they highlight that the answer types require models to perform tasks such as answer span extraction, candidate answer verification, and information aggregation based on relevance.

## 3 DetLLM: Diverse Retrieval Tool Augmented LLM

We now introduce our diverse retrieval tool augmented LLM (DetLLM) and show its promising capability on the proposed DivKnowQA benchmark by unifying the retrieval ability from the structured and unstructured knowledge sources.

To tackle a complex question, we follow the chain-of-thought (CoT) framework Wei et al. (2022) to decompose a complex question into single-hop questions where each single-hop question is knowledge-intensive, requiring supportive fact retrieval from a knowledge source.

We design a retrieval tool capable of retrieving from heterogeneous knowledge sources. For unstructured text knowledge, a dense passage retriever Izacard et al. (2022) is employed to retrieve relevant passages. For structured knowledge, we consider two modalities of structured knowledge to maximize the relevant information coverage. First, we transform the structured data into text passages by linearizing the relation triplets into passages in which case a sparse text retriever can be used to detect similar sources. Second, we propose a symbolic query generation module to map a natural language query to a structured query (e.g., SPARQL) to directly query against the KB (e.g., Wikidata). The benefits are twofold: (1) pinpointing precise knowledge, and (2) leveraging the compositionality of the query language and reducing the mere reliance on the language model's reasoning responsibility. Appendix A.8 shows the DetLLM flow for querying an LLM.

### 3.1 Question Decomposition and Planning

Our approach to answering a complex multi-hop question is inspired by the conceptual framework of DSP (Khattab et al., 2022). When dealing with a question that involves $n$ hops, we query the LLM $n$ times to generate retrieval queries and retrieve information from a knowledge source. The final query is used to ultimately arrive at the final answer, utilizing the retrieved passage to answer the last single-hop question. This process results in a total of $n + 1$ interactions with the LLM. At the

Table 2: Answer and Sub-Step Retrieval Accuracy on DIVKNOWQA.

|                | EM   | F1   | Recall | H1-R | H2-R |
|----------------|------|------|--------|------|------|
| Vanilla Prompt | 26.0 | 28.3 | 26.8   | 42.2 | -    |
| ReAct          | 16.1 | 18.4 | 19.0   | -    | -    |
| DSP            | 27.9 | 31.0 | 31.2   | 57.6 | 41.2 |
| DETLLM (our)   | **32.1** | **35.7** | **35.6** | **70.1** | **47.1** |

$j$-th LLM prompting, the LLM's task is to utilize the previously retrieved information to answer the $j-1$ single-hop question. It then dissects the original question $Q$ into the $j$-th subsequent single-hop questions $q_j$, which serve as a retriever query to gather information from a knowledge source.

## 3.2 MULTI-SOURCE KNOWLEDGE RETRIEVAL

In addressing the single-hop questions, our approach entails searching across diverse knowledge sources to gather supporting facts. To answer the subsequent single-hop question $q_j$, we begin by having the LLM generate semantically diverse queries, denoted as $Query_j = \{query_1^j, \ldots, query_t^j\}$. We set the LLM decoding temperature to $0.7$ to sample diverse queries.

In our approach, we treat unstructured and structured knowledge separately and retrieve relevant information from both knowledge sources. As mentioned, for unstructured knowledge, we use a dense retriever Contriever Izacard et al. (2022) to retrieve relevant passages, while for structured knowledge, we retrieve relevant information from both textual and structured formats. The preparation of the textual knowledge base involves linearizing KB triplets $(sub, relation, obj)$ into a string format "`sub relation obj`" after which we create a retrieval index for efficient passage retrieval using a sparse text retriever BM25 Robertson et al. (2009). The Contriever, trained on natural language corpus, is adaptable to unstructured knowledge but struggles when faced with linearized structured knowledge because it lacks natural language formatting. In contrast, the sparse retriever BM25 performs better with structured knowledge by using a keyword-based search methodology. We show the ablation study in Section 4.3.

In addition, we generate SPARQL queries to execute against the Wikidata engine to retrieve further relevant information. Our retrieval tool thus comprises three components: a sparse retriever, a dense retriever, and a symbolic query language generation module. These elements collectively enable the comprehensive retrieval of information from heterogeneous knowledge sources.

## 3.3 MULTI-SOURCE KNOWLEDGE RANKING

To consolidate the retrieved information obtained from the tool, we perform a ranking of information from various knowledge sources. The goal is to select the top-$k$ most relevant pieces of information. This selection is necessary because of the inherent length constraint of the language model, which prevents us from incorporating all the retrieved information into the prompt. To achieve this ranking, our approach leverages the off-of-shelf cross-encoder model Reimers and Gurevych (2019) to assess the relevance of each piece of retrieved information in the context of a single-hop question. We use the `sentence-transformers` package implementation with the model checkpoint `cross-encoder/ms-marco-MiniLM-L-6 -v2`.[1]

## 4 BENCHMARKING

### 4.1 EXPERIMENTAL SETUP

**Baselines** (1) ChatGPT (OpenAI, 2023): We employ OpenAI's ChatGPT model (`gpt-3.5-turbo`) by single-step query inputting the question and retrieved-context and

---

[1] `sentence-transformers`: https://www.sbert.net/

Table 3: Ablation Study on the Retrieval Strategy.

|  | EM | F1 | Recall | H1-R | H2-R |
|---|---|---|---|---|---|
| *w/o SPARQL* | | | | | |
| Text-KB(Sparse) | 27.9 | 31.0 | 31.2 | 57.6 | 41.2 |
| Text-KB(Dense) | 22.7 | 26.1 | 26.9 | 54.9 | 32.0 |
| Text(Sparse)-KB(Sparse) | 26.4 | 29.8 | 30.2 | 60.0 | 41.2 |
| Text(Dense)-KB(Sparse) | 30.7 | 35.0 | 35.5 | 68.9 | 46.8 |
| *w/ SPARQL* | | | | | |
| Text-KB(Sparse) | 28.8 | 31.9 | 32.9 | 58.0 | 42.9 |
| Text-KB(Dense) | 31.2 | 34.7 | 35.9 | 64.3 | 42.6 |
| Text(Sparse)-KB(Sparse) | 28.5 | 31.8 | 32.0 | 61.5 | 42.1 |
| **DETLLM (our)** | **32.1** | **35.7** | **35.6** | **70.1** | **47.1** |

obtaining its response as the final answer. (2) DSP (Khattab et al., 2022): We apply the demonstrate-search-predict framework to iteratively address complex QA tasks with the assistance of retrieved context. (3) ReAct (Yao et al., 2023): It leverages a synergistic approach, combining reasoning with tool usage. It involves verbally generating a reasoning trace and issuing the necessary commands to invoke a tool, which then takes action accordingly. We use `gpt-3.5-turbo` as the backbone model.

**Evaluation Metrics** To assess the accuracy and relevance of various models for factoid questions, we rely on established metrics. We report the exact match and F1 score for final answer quality, following the methodology of Yang et al. (2018). Besides, we report the Recall score indicating whether the ground-truth answer is a substring in prediction since LLM may generate extra information. In addition, we report the retrieval accuracy for each decomposed single-hop question denoted as H1-R and H1-2 for the two-hop question.

**Implementation Details** To ensure a comprehensive and equitable comparison, we offer baseline model access to both structured knowledge and unstructured knowledge as retrieval sources. In the case of the baseline model, the KB is converted into linearized passages, which are then combined with the unstructured knowledge, creating a unified source for retrieval. We use BM25 (Robertson et al., 2009) and Contriever (Izacard et al., 2022) as sparse and dense retrieval tools respectively. Unless specified otherwise, we experiment with a few-shot prompt that includes three human-annotated demonstrations along with task instructions to guide the model generation process.

## 4.2 MAIN RESULTS

**Comparing with State-of-the-Art LLMs** Table 2 presents the model performance results on the DIVKNOWQA. ReAct exhibits lower performance compared to the Vanilla prompt. The retrieval tool created for ReAct is specialized for querying unstructured knowledge. As the presence of irrelevant passages distracts the LLM Chen et al. (2023); Mallen et al. (2023), the iterative reasoning accumulates errors, leading to less accurate answers. Conversely, DSP outperforms both Vanilla Prompt and ReAct, thanks to its robust search module designed to engage with frozen retrievers. DSP enhances a single retrieval query into multiple queries, employing a fusion function to rank candidate passages and identify the most relevant one. However, the search module cannot effectively retrieve structured knowledge. Our model stands out as the top-performing model, demonstrating its capability to generate symbolic language for retrieval from structured knowledge.

**Retrieval Performance** Table 2 also presents the single-step retrieval accuracy. Among the baseline methods, comparing single-step generation e.g. Vanilla Prompt with the multi-step generation e.g. ReAct and DSP, the retrieval accuracy increases due to the decomposed query from the multi-step generation process. On the other hand, the DETLLM shows stronger retrieval performance compared to DSP due to the careful retrieval tool design, the unstructured and structured knowledge is treated separately. This finding underscores the importance of having a robust retrieval strategy to provide reliable and focused information, grounding the LLM on relevant supportive facts.

Table 4: Breakdown Analysis on SPARQL generation.

|  | QID | QID+REL | QID* |
|---|---|---|---|
| Text-KB(Sparse) | 26.5 | 22.4 | 6.91 |
| Text-KB(Dense) | 31.8 | 27.6 | 7.87 |
| Text(Sparse)-KB(Sparse) | 26.3 | 22.6 | 7.34 |
| Text(Dense)-KB(Sparse) | 29.7 | 26.5 | 7.66 |

Table 5: Experiment results using Oracle knowledge source retrieval in each sub-step.

|  | EM | F1 | Recall | H1-R | H2-R |
|---|---|---|---|---|---|
| Oracle_Text | 26.8 | 31.3 | 33.4 | 96.4 | 51.7 |
| Oracle_KB | 38.1 | 40.0 | 42.2 | 100.0 | 62.1 |
| Oracle_All | 48.7 | 52.2 | 52.8 | 100.0 | 96.7 |

## 4.3 DISCUSSION

**Ablation Study** Table 3 presents the results of an ablation study involving three key factors: a) the integration of heterogeneous knowledge sources, b) the choice between dense and sparse retrievers, and c) the incorporation of SPARQL. Our findings indicate that optimal performance is achieved when handling heterogeneous knowledge sources separately, combined with careful retriever tool selection. The unsupervised dense retriever (i.e., Contriever), trained on natural language corpus, demonstrates adaptability to unstructured knowledge but loses its advantage when dealing with linearized structured knowledge due to the absence of natural language formatting. Conversely, the sparse retriever BM25 performs better on structured knowledge, relying on keyword-based search methodologies. Furthermore, the SPARQL tool consistently outperforms its counterparts in all settings, showcasing improvements regardless of the integration of knowledge sources and the choice of retriever.

**SPARQL Generation Analysis** Symbolic language generation is an essential tool, which is executed against the Wikidata engine to assist with structured knowledge retrieval. We provide a detailed breakdown analysis of SPARQL generation in Table 4. "QID" represents the percentage of examples with entity IDs correctly linked to Wikidata. Additionally, we present the percentage of examples linked to the Wikidata in terms of both entity IDs and relation IDs denoted as "QID+REL". The last column, labeled "QID*", showcases the percentage of examples with great potential for accurate identification through entity disambiguation. In our experimental process, we first identify the entity name from the decomposed question as a retriever query and then link the entity from the query to Wikidata. The returned results provide a list of candidate Wikidata entities, from which we select the most semantically similar one by computing the similarity between the query and the entity's description. The displayed number reveals that this heuristic entity disambiguation process fails to recognize those examples that actually contain the correct entity ID within the candidate list. This highlights a potential avenue for further improving model performance.

**Establishing Oracle Performance** In Table 5, we present the experimental results obtained using Oracle information. In these experiments, we grant the model access to ground-truth passages from the Oracle Text and linearized KB triplets from the KB Oracle. A notable observation is the comparison between Text Oracle and KB Oracle. We find that KB Oracle exerts a more significant influence on the final results. This is because structured knowledge contains long-tail knowledge, showing the necessity to effectively explore structured knowledge. Furthermore, when both Text and KB Oracle sources are provided, the model's performance reaches an Exact Match (EM) rate of 48.7%, highlighting the necessity of each knowledge source. In comparison to our current established results from DETLLM, this benchmark reveals substantial room for the research community to further explore and improve upon.

**Comparing with Closed-book LLM** Table 6 presents a comparison between DETLLM and LLM performance in the closed-book setting, where no external knowledge is accessible. We demonstrate that DETLLM exhibits improvements in scenarios distinct from the closed-book setting. We observe

Table 6: Comparison between the closed book setting and open domain retrieval.

|  | EM | F1 | Recall | H1-R | H2-R |
|---|---|---|---|---|---|
| Closed Book | 30.2 | 33.8 | 31.2 | - | - |
| DETLLM | 32.1 | 35.7 | 35.6 | 70.1 | 47.1 |

that only 50.8% of examples answered correctly by our DETLLM are also present in the closed-book setting, highlighting the orthogonal performance of DETLLM compared to the closed-book setting. The combination of correctly answered examples accounts for 45.4% of the entire dataset. One plausible hypothesis is that the closed book setting enables the LLM to access knowledge stored in its memory, reducing the impact of retriever errors. We also suggest a potential research direction, which involves designing a strategy to switch between the closed book setting and open domain retrieval to achieve optimal performance.

## 5 RELATED WORK

### 5.1 ASSESSING THE REASONING ABILITY OF LLMS

LLMs (Brown et al., 2020; Touvron et al., 2023; Nijkamp et al., 2023) have exhibited notable advancements in their capabilities, particularly in the domain of reasoning skills. These skills encompass various categories, including inductive reasoning (Wang et al., 2023; Yang et al., 2022), deductive reasoning (Creswell et al., 2023; Han et al., 2022), and abductive reasoning (Wiegreffe et al., 2022; Lampinen et al., 2022), depending on the type of reasoning involved. Current research efforts have predominantly focused on evaluating LLMs in the context of open-ended multi-hop deductive reasoning. These scenarios involve complex question-answering tasks (Yang et al., 2018; Gu et al., 2021; Trivedi et al., 2022b; Liu et al., 2023b) and fact-checking (Jiang et al., 2020). Notably, our work contributes to this landscape by introducing an additional layer of complexity: the integration of multi-hop and multi-source reasoning. In our approach, we retrieve supporting facts from heterogeneous knowledge sources, further enhancing the challenges posed to LLMs in this deductive reasoning context.

### 5.2 RETRIEVAL-AUGMENTED LLMS

Retrieval-Augmented Large Language Models (RALLMs) are semi-parametric models that integrate both model parameters and a non-parametric datastore to make predictions. RALLMs enhance LLMs by updating their knowledge (Izacard et al., 2023; Khandelwal et al., 2020; Yavuz et al., 2022; Mallen et al., 2023), providing citations to support trustworthy conclusions (Menick et al., 2022; Gao et al., 2023). RALLMs can retrieve information in an end-to-end fashion within a latent space (Khandelwal et al., 2020; 2021; Min et al., 2023), or they can follow the retrieve-then-read paradigm, leveraging an external retriever to extract information from textual sources (Ram et al., 2023; Khattab et al., 2022). Our approach adheres to the retrieve-then-read paradigm, with a specific emphasis on multi-source retrieval, advocating for structured knowledge retrieval through symbolic generation.

## 6 CONCLUSION

We introduce the DIVKNOWQA, designed to evaluate the proficiency of question-answering systems, especially those enhanced by retrieval tools, in addressing knowledge-intensive questions with a strong emphasis on multi-hop multi-source retrieval. This dataset is constructed through automated data generation and subsequent human verification, minimizing manual effort. Our evaluation encompasses both standard LLMs and LLMs augmented with retrieval tools. Notably, we identify that this task presents a new challenge for state-of-the-art models due to the demand for structured knowledge retrieval and the inherent lack of prior knowledge in this context. To tackle this challenge, we propose the DETLLM, which incorporates diverse retrieval tools including innovative symbolic query generation for retrieving information from the structured knowledge source. In the future, we are keen on enhancing LLMs' capabilities in understanding and generating symbolic language, as well as exploring methods to improve performance on knowledge-intensive and complex question-answering tasks.

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

## A  APPENDIX

### A.1  SINGLE-HOP KNOWLEDGE BASE QUESTION GENERATION PROMPT

Prompt 1: Single-Hop Knowledge Base Question Generation

```
Instruction: Question generation given the following information:
1) Answer
2) Short relation between the question entity and the answer
3) Question entity.

IMPORTANT: The answer must be avoided in the question.

Answer: Jacques Boigelot;
Relation: director;
Question Entity: Peace in the Fields;
Question: Who directs Peace in the Fields?

Answer: Academy Award for Best Sound Mixing;
Relation: award received;
Question Entity: Douglas Shearer;
Question: Which award does Douglas Shearer receive?

Answer: Rio de Janeiro;
Relation: place of birth;
Question Entity: David Resnick;
Question: Where was David Resnick born?
```

### A.2  MULTI-HOP COMPLEX QUESTION GENERATION PROMPT

Prompt 2: Multi-Hop Complex Question Generation

```
Instruction: Compose 2 single-hop questions into a 2-hop question given:
1) Hop1 question
2) Hop1 answer
3) Hop2 question.

Hop1 question: Who said a rose by any other name would smell just as
    sweet?
Hop1 answer: Juliet
Hop2 question: What is the cause of death of Juliet?
Composed question: What is the cause of death of the person who said a
    rose by any other name would smell just as sweet?

Hop1 question: Who hosted The Price Is Right before Bob Barker?
Hop1 answer: Bill Cullen
Hop2 question: What is the medical condition of Bill Cullen?
Composed question: What is the medical condition of the person who
    hosted The Price Is Right before Bob Barker?

Hop1 question: Who wrote If You Go Away on a Summer's Day?
Hop1 answer: Rod McKuen
Hop2 question: Which record company does Rod McKuen own?
Composed question: Which record company does the person who wrote If You
    Go Away on a Summer's Day own?
```

### A.3  BENCHMARK PROMPT

To use the DETLLM method and generate the final answer, three steps are followed: (1) First-hop prompting, (2) Second-hop prompting, and (3) Final answer generation. The prompt for each stage is provided below. For simplicity, we denote the $k$ retrieved passages as "*Context: [[1] ... [k]]*".

Prompt 3: First Hop

```
Write a search query, query entity, and SPARQL that will help answer a
    complex question.
Follow the following format.
Context: ${sources that may contain relevant content}
Question: ${the question to be answered}
Rationale: Let's think step by step. Based on the context, we have
    learned the following. ${information from the context that provides
    useful clues}
Search Query: ${a simple question for seeking the missing information}
Query Entity: ${query entity name from search query}
SPARQL: ${SPARQL query used to query against Wikidata}

Example 1
Context:
Question: What are the occupations of the person who holds the most
    women's Wimbledon titles?
Rationale: Let's think step by step. Based on the context, we have
    learned the following. Decompose the question to answer the
    following single-hop questions. 1. Who holds the most women's
    Wimbledon titles? 2. What are the occupations of this person
Search Query: Who holds the most women's Wimbledon titles?
Query Entity: women's Wimbledon titles
SPARQL: None

Example 2
Context:
Question: Which bay is the name of David Resnick's place of birth?
Rationale: Let's think step by step. Based on the context, we have
    learned the following. Decompose the question to answer the
    following single-hop questions. 1. Where was David Resnick born? 2.
    Which bay is the name of this place
Search Query: Where was David Resnick born?
Query Entity: David Resnick
SPARQL: SELECT ?place WHERE {wd:Q962183 wdt:P19 ?place.}

Example 3
Context:
Question: Is the person who directed the film The Shape of Water a
    member of the Writers Guild of America, West?
Rationale: Let's think step by step. Based on the context, we have
    learned the following. Decompose the question to answer the
    following single-hop questions. 1. Who directed the film the shape
    of water? 2. Is the person the person a member of the Writers Guild
    of America, West?
Search Query: The director of the film The Shape of Water
Query Entity: The Shape of Water
SPARQL: SELECT ?name WHERE {wd:Q26698156 wdt:P57 ?name.}

Target Question
Context:
Question: How many organizations is the 26th president of the United
    States a member of?
Rationale: Let's think step by step. Based on the context, we have
    learned the following. Decompose the question to answer the
    following single-hop questions. 1. who is the 26th president of the
    United States? 2. How many organizations is this person a member of?
Search Query: 26th president of the United States
Query Entity: None
SPARQL: None
```

Prompt 4: Second Hop

```
Write a search query, query entity, and SPARQL that will help answer a
    complex question.
```

```
Follow the following format.
Context:${sources that may contain relevant content}
Question: ${the question to be answered}
Rationale: Let's think step by step. Based on the context, we have
    learned the following. ${information from the context that provides
    useful clues}
Search Query: ${a simple question for seeking the missing information}
Query Entity: ${query entity name from search query}
SPARQL: ${SPARQL query used to query against Wikidata}

Example 1
Context:[[1] ... [k]]
Question: What are the occupations of the person who holds the most
    women's Wimbledon titles?
Rationale: Let's think step by step. Based on the context, we have
    learned the following. Wimbledon is a tennis tournament, and tennis
    player Martina Navratilova holds the most women's Wimbledon titles.
    The second step is to answer what are the occupations of this person.
Search Query: What are the occupations of Martina Navratilova?
Query Entity: Martina Navratilova
SPARQL: SELECT ?name WHERE {wd:Q54545 wdt:P106 ?name.}

Example 2
Context:[[1] ... [k]]
Question: Which bay is the name of David Resnick's place of birth?
Rationale: Let's think step by step. Based on the context, we have
    learned the following. David Resnick was born in Rio de Janeiro. The
    second step is to answer which bay is the name of Rio de Janeiro?
Search Query: which bay is the name of Rio de Janeiro?
Query Entity: Rio de Janeiro
SPARQL: None

Example 3
Context:[[1] ... [k]]
Question: Is the person who directed the film The Shape of Water a
    member of the Writers Guild of America, West?
Rationale: Let's think step by step. Based on the context, we have
    learned the following. The Shape of Water is directed by Guillermo
    del Toro. The second step is to answer is the person a member of the
    Writers Guild of America, West
Search Query: the organization Guillermo del Toro is in
Query Entity: Guillermo del Toro
SPARQL: SELECT ?name WHERE {wd:Q219124 wdt:P463 ?name.}

Target Question
Context:[[1] ... [k]]
Question: How many organizations is the 26th president of the United
    States a member of?
Rationale: Let's think step by step. Based on the context, we have
    learned the following. The 26th president of the United States is
    Theodore Roosevelt. The second step is to answer how many
    organizations he is a member of.
Search Query: How many organizations is Theodore Roosevelt a member of?
Query Entity: Theodore Roosevelt
SPARQL : SELECT (COUNT(?organization) as ?count) WHERE { wd:Q33866
    wdt:P463 ?organization. }
```

Prompt 5: Final QA Step

```
Answer questions with short factoid answers.
Follow the following format.
Context:${sources that may contain relevant content}
Question: ${the question to be answered}
```

```
Rationale: Let's think step by step. ${a step-by-step deduction that
    identifies the correct response, which will be provided below}
Answer: ${a short factoid answer, often between 1 and 5 words}

Example 1
Context: [[1] ... [k]]
Question: What are the occupations of the person who holds the most
    women's Wimbledon titles?
Rationale: Let's think step by step. Martina Navratilova is a tennis
    player, writer, novelist, and autobiographer.
Answer: tennis player, writer, novelist, and autobiographer

Example 2
Context: [[1] ... [k]]
Question: Which bay is the name of David Resnick's place of birth?
Rationale: Let's think step by step. David Resnick was born in Rio de
    Janeiro, and "Rio de Janeiro" was the name of Guanabara Bay.
Answer: Guanabara Bay

Example 3
Context:[[1] ... [k]]
Question: Is the person who directed the film The Shape of Water a
    member of the Writers Guild of America, West?
Rationale: Let's think step by step. Guillermo del Toro Gomez is a
    filmmaker, he is a member of the Writers Guild of America, West.
Answer: yes

Target
Context:[[1] ... [k]]
Question: How many organizations is the 26th president of the United
    States a member of?
Rationale: The 26th president of the United States was Theodore
    Roosevelt. He is a member of 5 organizations.
Answer: 5
```

## A.4 HUMAN ANNOTATION INSTRUCTION

We show the instructions and annotating examples provided to human annotators to annotate the dataset as below.

**Overall Instruction** The goal of the annotation is to judge and revise the complex question chained by two single-hop questions. To complete this goal, you need to do the following two tasks:

- Judge and revise the single-hop question generated from the knowledge base triplet.
- Judge and revise the composed complex question.

**Task 1** Given a triplet (subject, relation, object) and a machine-generated question as shown below, you need to judge the quality of the generated question and whether it is acceptable, needs revision, or is rejected. If the question can be revised, please revise the question rather than reject it. If the question is too poor to revise, reject the question.

```
Triplet: (LeBron James; child; [Bryce James, Zhuri James, Bronny James])
Question: How many children does LeBron James have?
```

An accepted triplet question should satisfy the following criteria:

- The question focuses on the subject w.r.t relation.
- The question should sound natural and fluent.
- The answer to the generated question should be the object, thus the object cannot be shown in the question.

**Task 2** Judge and revise the composed complex question given the following information. If the question can be revised, please revise the question rather than reject it. If the question is too poor to revise, reject the question and choose the reason for rejection.

Below is a list of provided information:

- Two single-hop question-answer pairs: "(Question 1, Answer 1)" and "(Question 2, Answer 2)".
- The bridging entity "Bridge Entity" that chains two single-hop questions together.
- Machine generated composed question "Composed Question".

```
Question 1: Who is the highest-paid athlete in the NBA
Answer 1: LeBron James
Question 2: How many children does LeBron James have?
Answer 2: [3, three]
Bridge Entity: LeBron James
Composed Question: How many children does the highest-paid athlete in
    the NBA have?
```

An accepted question should meet the following criteria:

- The composed question must be constructed using two single-hop questions, with the answer to the first question becoming the subject of the second question.
- Ensure that the composed question does not reveal the answer itself.
- Use 'Answer 2' as the answer to the composed question.

If you choose to reject the question, please select one of the following reasons. If your reason is not listed, choose 'Other' and include a comment.

- Circular question: Two single-hop questions are the same question.
- Bridge entity answer leaking.
- Final answer leaking.
- Change in the original meaning of single-hop questions.
- Other.

A.5 AN EXAMPLE OF THE TWO-HOP MULTI-SOURCE QUESTIONS IN DIVKNOWQA

**Q:** How many awards has the first person to walk on the moon received?

**A**: 26

**Multi-Hop Multi-Source Reasoning**

Q1: **Who was the first person to walk on the moon?**

A1: Neil Armstrong (Supporting facts: [1], [2])

Q2: **How many awards has Neil Armstrong received?**

A2: 26  (Supporting fact: [3])

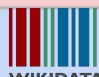

**Unstructured Knowledge**

Paragraph A, Moon landing

[1] *This was accomplished with two US pilot-astronauts flying a Lunar Module on each of six NASA missions across a 41-month period starting on 20 July 1969 UTC, with **Neil Armstrong** and…*

Paragraph B, Purdue University

[2] ***Neil Armstrong** (the first person to walk on the moon)*

**Structured Knowledge**

[3] [*"Presidential Medal of Freedom","Order of the White Elephant",  "Cullum Geographical Medal", "National Aviation Hall of Fame",* …] (In total 26 items)

```
SPARQL Query:
SELECT (COUNT(?item) AS ?count)
WHERE {wd:Q1615 wdt:P166 ?item.}
```

Figure 2: An example of the two-hop multi-source questions in our dataset.

## A.6 ANALYSIS OF DIVKNOWQA

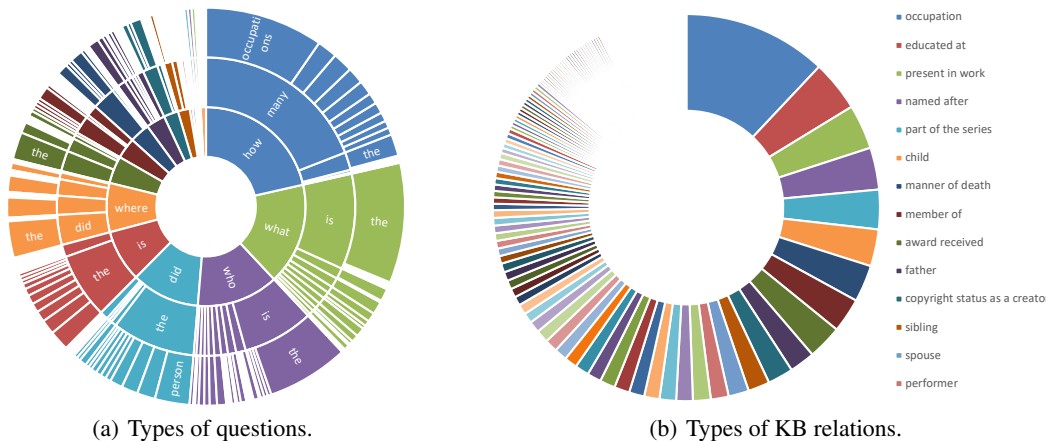

(a) Types of questions.

(b) Types of KB relations.

Figure 3: Types of (a) questions, and (b) KB relations, covered in DIVKNOWQA.

## A.7 ANECDOTAL EXAMPLES FOR DIVKNOWQA

Table 7: Types of multi-hop reasoning required to answer questions in DIVKNOWQA. Two single-hop questions are shown: TextQA is sampled from NQ, and KBQA is generated using the sampled KB-Triplet. The question from **DIVKNOWQA** is based on those two single-hop questions.

| Order | Type | % | Example |
|---|---|---|---|
| **Text → KB** | short entity | 20.3 | TextQA: Who is Rafael Nadal married to?
Answer: María Francisca Perelló
KB-Triplet: (Rafael Nadal, spouse, María Francisca Perelló)
KBQA: Who won the Men's US Open 2017?
Answer: Rafael Nadal
**DIVKNOWQA**: Who is the person married to the winner of the Men's US Open 2017?
**Answer**: María Francisca Perelló |
| | yes/no | 17.9 | TextQA: Who sang When the Lights Went Out in Georgia?
Answer: Vicki Lawrence
KB-Triplet: (Vicki Lawrence, hair color, red hair)
KBQA: What is Vicki Lawrence's hair color?
Answer: red hair
**DIVKNOWQA**: Is the hair color of the singer of "When the Lights Went Out in Georgia" gray?
**Answer**: no |
| | aggregate | 21.1 | TextQA: who does Meg 's voice on Family Guy?
Answer: Vicki Lawrence
KB-Triplet: (Mila Kunis, child, [Wyatt Kutcher, Dimitri Kutcher])
KBQA: How many children does Mila Kunis have?
Answer: Two
**DIVKNOWQA**: How many children does the person who does Meg's voice on Family Guy have?
**Answer**: Two |
| **KB → Text** | short entity | 20.7 | KB-Triplet: (William Weatherall Wilkins, present in work, Mary Poppins Returns)
KBQA: In which work is William Weatherall Wilkins present?
Answer: Mary Poppins Returns
TextQA: Who play Mary Poppins in Mary Poppins Returns?
Answer: Emily Blunt
**DIVKNOWQA**: Who plays Mary Poppins in the work in which William Weatherall Wilkins is present?
**Answer**: Emily Blunt |
| | yes/no | 20.0 | KB-Triplet: (Girl #2, present in work, High School Musical)
KBQA: In which work is Girl #2 present?
Answer: High School Musical
TextQA: What grade were they in in high school musical 1?
Answer: juniors
**DIVKNOWQA**: Were they seniors in the work in which Girl #2 is present?
**Answer**: no |

## A.8 THE ILLUSTRATION OF DETLLM

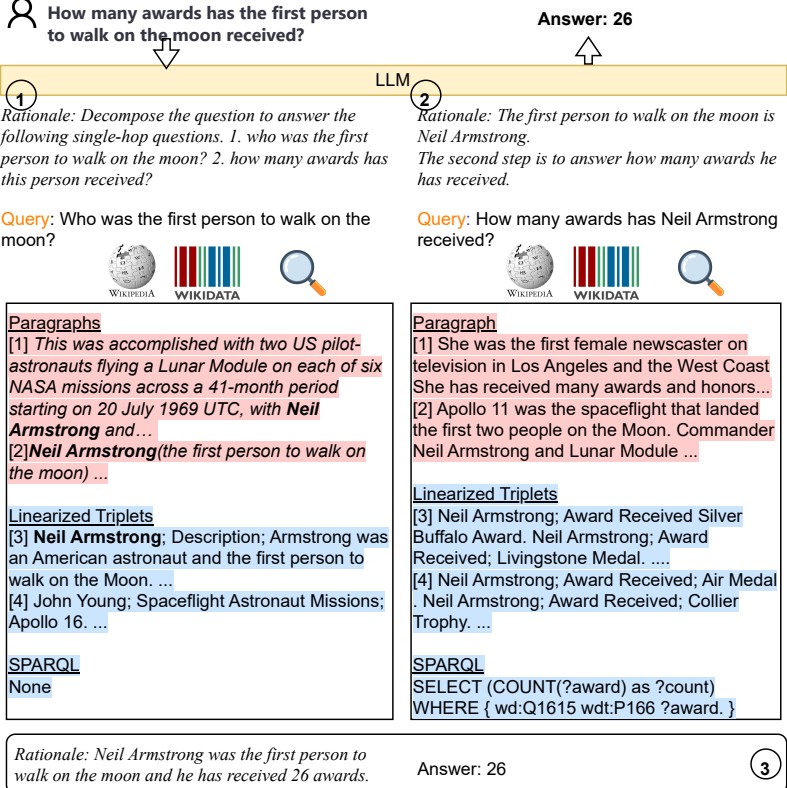

Figure 4: The illustration of DETLLM to instruct LLMs for addressing multi-source multi-hop questions.

