# OpenReview forum: "DivKnowQA: Verifying the Reasoning Ability of LLM Through Open-Domain Question Answering Over Knowledge Base and Text"
_ICLR.cc/2024/Conference — ICLR 2024 Conference Withdrawn Submission_

### Official Review · Reviewer_r9Z6 · 2023-10-18

**Soundness:** 2 fair
**Presentation:** 2 fair
**Contribution:** 2 fair
**Rating:** 3
**Confidence:** 4

**Summary:**

This paper introduces DivKnowQA, a benchmark for open-domain question answering consisting of 940 human-annotated questions. These questions are sourced from external knowledge repositories like WikiData,  requiring models to use external information and engage in multi-step reasoning to provide answers. The dataset encompasses a wide range of domains, making it challenging for models to leverage external structured data effectively. In addition to presenting this dataset, the paper introduces a method called Diverse Retrieval Toolbox  Augmented LLM, abbreviated as DetLLM. DetLLM employs a textual passage retriever and a symbolic generation tool to supply external knowledge to the model, resulting in superior performance compared to other baseline methods.

**Strengths:**

As outlined in the summary, this paper makes a dual contribution. First, it introduces a dataset called DivKnowQA, which stands out for its diversity in question types spanning various domains. These questions necessitate multi-hop reasoning and typically require external knowledge to answer.

Additionally, alongside the dataset, the paper introduces DetLLM, a method based on a retrieval system and symbolic textual generation tools. This method has proven to be effective in harnessing structured knowledge to enhance the performance of Language Model Models (LLMs), outperforming the baseline methods.

**Weaknesses:**

1. Dataset Quality: A significant limitation of this paper pertains to the quality of the proposed dataset. DivKnowQA comprises only 940  question-answer pairs, a relatively small quantity even for benchmarking purposes. Moreover, because of the scarcity, the diversity of the questions is in doubt. Lastly, although these pairs undergo human verification, they originate from machine-generated sources, thus the questions may not be natural.  The authors could enhance the paper by including a table that compares the size of DivKnowQA with similar datasets, helping practitioners determine whether this dataset suits their needs.
2. Novelty of the Construction Process: The methodology employed to create  DivKnowQA bears a striking resemblance to that used for HotpotQA,  another dataset built on knowledge graphs and designed for multi-hop reasoning. Moreover, the analysis of the dataset in this paper is similar to that in HotpotQA, e.g. Figure 2 of HotpotQA and Figure 2 of DivKnowQA. Given that HotpotQA is substantially larger and was extensively studied, it is important for the authors to elaborate on the distinctive aspects that set DivKnowQA apart from HotpotQA.
3. Models being tested: If DivKnowQA is supposed to be a benchmark, then at least more than 1 LLM should be used. In this paper, only ChatGPT 3.5 is tested.

**Questions:**

Some notes in the paper draft are not removed for the submitted version. E.g. in page 7, "*here insert the dataset name*" should be replaced with DivKnowQA.  In page 8, "referred  to as *approach*" should be replaced with "referred to as *DetLLM*".

---

> ### Author Response · Authors · 2023-11-23
> **Response to Reviewer r9Z6**
>
> Thank you for your comprehensive review of our submission. We appreciate your detailed analysis and have taken your feedback into consideration to improve our manuscript. Below, we address each of the concerns and questions you raised.
>
> ### Dataset Quality
>
> **Weakness 1:**
> Enhance the paper by including a table that compares the size of DivKnowQA with similar datasets, helping practitioners determine whether this dataset suits their needs.
>
> **Response:**
> Our dataset, DivKnowQA, has a unique focus on evaluating the reasoning capabilities of Language Models (LLMs) rather than serving as a training resource. This distinction is important as it shapes the composition and size of our dataset. While traditional datasets are typically large enough to facilitate training, DivKnowQA provides a human-annotated test set specifically designed for assessment purposes.
>
> To clarify this difference and help practitioners understand how DivKnowQA fits into the broader landscape, we have included a table in our paper that compares the sizes and purposes of similar datasets. This comparison highlights DivKnowQA's role as a specialized tool for evaluating LLM reasoning abilities, as opposed to being a general training dataset. The table will provide a clear perspective on how our dataset differs in terms of size and focus compared to others in the field.
>
> **Table 1:** Comparative Statistics of Training-Oriented Datasets in Language Model Evaluation
>
> | Dataset         | Train   | Dev    | Test  |
> |-----------------|---------|--------|-------|
> | HybridQA        | 62,682  | 3,466  | 3,463 |
> | OTTQA           | 41,469  | 2,214  | 2,158 |
> | NQ-TABLES       | 9,594   | 1,068  | 966   |
> | TAT-QA          | 13,241  | 1,655  | 1,655 |
> | MULTIMODELQA    | 23,817  | 2,441  | 3,660 |
> | ManymodelQA     | 2,036   | 3,055  | 5,099 |
> | FinQA           | 6,251   | 883    | 1,147 |
> | HetpQA          | 3,818   | 424    | 2,319 |
> | CompMix         | 4,966   | 1,680  | 2,764 |
> | WikiMovies-10K  | 96k     | 10k    | 10k   |
> | MetaQA          | 329,282 | 39,138 | 39,093 |
>
> **Weakness 2:**
> Elaborate on the distinctive aspects that set DivKnowQA apart from HotpotQA.
>
> **Response:**
> Our dataset, DivKnowQA, distinguishes itself from HotpotQA in two key aspects. Firstly, while HotpotQA constructs its multi-hop questions primarily using Wikipedia as a singular knowledge source, DivKnowQA extends beyond textual corpora to incorporate structured knowledge bases (KBs). This approach allows for multi-source, multi-step reasoning, focusing on retrieving long-tail knowledge and KB triplets not typically found in text corpora. Secondly, DivKnowQA emphasizes the importance of generating structured queries specifically tailored for KB retrieval. This is in contrast to HotpotQA, where the focus is on aggregation questions based on text passages. Our approach enables a more intricate exploration of structured data, which is pivotal in handling complex query scenarios not covered by HotpotQA.
>
> **Weakness 3:**
> If DivKnowQA is supposed to be a benchmark, then at least more than 1 LLM should be used. In this paper, only ChatGPT 3.5 is tested.
>
> **Response:**
> We selected ChatGPT 3.5 as our primary LLM for testing due to its advanced capabilities. Our focus was to benchmark the performance of various tool-augmented LLMs against ChatGPT 3.5's backbone. This comparison aims to demonstrate that even with state-of-the-art models like ChatGPT 3.5, there's room for improvement in handling diverse, heterogeneous knowledge sources through retrieval augmentation. In our upcoming paper revisions, we plan to include additional experiments using GPT-4 as the backbone. However, due to time constraints, these in-depth analyses will be part of our future work.
>
> **Question:**
> Some notes in the paper draft are not removed for the submitted version. E.g. in page 7, "here insert the dataset name" should be replaced with DivKnowQA. In page 8, "referred to as approach" should be replaced with "referred to as DetLLM".
>
> **Response:**
> We have carefully reviewed the entire manuscript and removed all draft notes, ensuring a polished final version. The specific instances you mentioned on pages 7 and 8 have been corrected, along with a thorough check for any similar oversights.

---

### Official Review · Reviewer_dqDr · 2023-11-01

**Soundness:** 3 good
**Presentation:** 3 good
**Contribution:** 2 fair
**Rating:** 5
**Confidence:** 3

**Summary:**

In this paper authors present a benchmark dataset that is Human verified that focuses on abilities of LLMs to use both the structured knowledge and textual sources. Benchmark is created keep in mind the importance is given to both structure source and textual source. It describes the process of dataset creation in detail. It also presents a baseline prompting  methods for LLMs that can aid them in solving this benchmark showing better results over existing LLM prompting methods.

**Strengths:**

Benchmark dataset that is carefully curated to test LLMs ability to use both structured and textual data.
Detailed analysis of the benchmark and process used in arriving it.

**Weaknesses:**

Evaluation seems to be somewhat flawed to me. Some of the baselines use only one of the two sources and then the proposed prompting method uses both of them together.
Is there way authors could have provided information from both sources to the baseline methods to be fair? Given the benchmark needs both sources to be used for answering, its not fair to cripple some of the baseline methods not providing all the data to generate answer.
Several details of how entity linking is performed etc are missing.

**Questions:**

Can you provide an example walk through of how DETLLM solves a question from the benchmark to make things clear on the overall process followed compared baselines.
Do you have results for both linearised KB triples and text provided to baseline methods by devising some prompt?

---

> ### Author Response · Authors · 2023-11-23
> **Response to Reviewer dqDr**
>
> Thank you for your thoughtful and detailed feedback on our submission. We have carefully considered your points and have made relevant revisions to our manuscript to address your concerns.
>
> ### Evaluation Methodology
>
> **Weakness 1a:**
> The evaluation methodology seemed somewhat flawed due to the usage of only one of the two sources (structured knowledge or textual sources) in some baseline methods, while our proposed method used both.
>
> **Response:**
> To address this concern, we have revised our evaluation strategy in the revised manuscript. We now provide information from both sources (structured knowledge and textual sources) to all baseline methods in Table 2. This ensures a fair and equitable comparison across all evaluated methods, as detailed in Section 4.2 of the revised paper.
>
> **Weakness 1b:**
> Several details of how entity linking is performed etc are missing.
>
> In the revised paper (refer to Section 3), we have elaborated on the instructions for query decomposition provided to the LLM, ensuring a more comprehensive understanding of the process.
>
> We use ChatGPT to translate a decomposed single-hop question to SPARQL. The prompt is shown in Appendix A.3. For the entity linking, we will first identify the entity named mentioned in the natural language query, then search for the candidate QID list from Wikidata API, and then compare the semantic similarity of each candidate item description with the natural language query. We will pick the most similar one as a linked entity.
>
> ### Questions
>
> **Question 1:**
>
> Can you provide an example walk through of how DETLLM solves a question from the benchmark to make things clear on the overall process followed compared baselines?
>
> **Response:**
> To enhance understanding, we have included a detailed prompt walkthrough of the overall process, especially focusing on how DETLLM solves a question from the benchmark in Appendix A.3 BENCHMARK PROMPT. In addition, we provide Figure 4 from Appendix A.8 as an illustration of DETLLM to instruct LLMs for addressing multi-source multi-hop questions.
>
> **Question 2:**
>
> Do you have results for both linearised KB triples and text provided to baseline methods by devising some prompts?
>
> **Response:**
> Yes, we updated the experiments results in Table 1 where all baseline models have access to both the textual and KB knowledge source. And in the 4.3 discussion, we present the results of an ablation study involving three key factors: a) the integration of heterogeneous knowledge sources, b) the choice between dense and sparse retrievers, and c) the incorporation of SPARQL. The ablation study shows the proposed setting is the optimal for handling multi-source multi-step QA task.

---

### Official Review · Reviewer_7HFZ · 2023-11-05

**Soundness:** 2 fair
**Presentation:** 2 fair
**Contribution:** 2 fair
**Rating:** 5
**Confidence:** 4

**Summary:**

This paper introduces a new QA dataset that jointly exploit open-domain question answering over structured KBs and unstructured text. The paper argues that even though there has been some work in developing QA datasets that exploit KBs and text, but more focus have been given to the textual component part. For example, no datasets come with ground-truth annotations of structured query language (such as SPARQL).  Such structured languages can be especially helpful for aggregation queries such as “How many European cities have a population > x?” as these are harder to answer from text but structured languages have operators and conditional statements which make it very easy.

The process of creating a multi-hop question begins with using an entity-linked single-hop question in the NaturalQuestions dataset along with its entity-linked answer. This is followed by gathering one-hop triples in the Wikidata knowledge graph where the question and answer entities are either subjects or objects. To ensure that KB reasoning is equally an important part of answering as text, they retain KB triples such that the corresponding entity is not present in Wikidata. Next, a single-hop question is generated from the triples which is followed by combining two single-hop questions (one from NQ and one from KB triple) into a multi-hop question. The method heavily relies on GPT-3.5 to both generate the question from KB as well as combining two question to from a multi-hop question. Apart from fact-based entity centric questions, yes/no as well as aggregate questions are generated (e.g. count-based questions). To ensure high quality of the dataset, each question is verified by two human experts and the resulting dataset consists of 940 questions.

The paper also contributes a model that employs question decomposition, retrieval from diverse knowledge sources to answer the question. GPT-3.5 is employed heavily in most steps.

**Strengths:**

**Originality**

* A dataset of multi-hop questions created by composing two simpler questions from different sources have been employed before. However, this paper does a decent job to ensure one of the entity is not popular enough to have a Wikipedia/Wikidata entry. Therefore the dataset contribution is original enough.

**Quality**

* The human verification is likely to ensure that the dataset is of reasonably high quality.

**Clarity**

* I found the dataset creation steps to be clearly understandable. The motivation of having such a dataset was clear. However, I found section 3 which explains the contributed model to miss many details (more in the weakness/question section). Similarly I found the experiment section to be very hastily written missing key details and leaving several important questions unanswered (more later). Therefore I believe the paper will significantly benefit from a round of re-writing.

**Significance**

* Even though the dataset is small (only 940 questions), the dataset has potential to be integrated into a larger benchmark for testing the reasoning ability of LLMs. However, I am really unclear about the effectiveness of the proposed model. This combined with several missing details make me question the significance of the paper in its current state.

**Weaknesses:**

* Even though I think the decision to include KB triples where the subject/object entity is not present in Wikidata to ensure equal reasoning importance to both KB and text is reasonable, but even though an entity is not captured in Wikidata, that doesnt necessarily mean that fact is not present in web-text. Since LLMs are trained on web-text, what is the guarantee that an LLM would require KB fact to answer the question? It is possible that the knowledge corresponding to the KB triplet is present in the LLM because it picked that up from a text snippet (not in Wikipedia) during pre-training.

* I think the model section (section 3) is missing many important details that severely hinder readability as well as trying to understand the limitations of the model. For example (and this is only a subset of missing information):

  * What are the instructions for query decomposition given to LLM?
  * How does the retriever work?
  * Similarly what is the model that generates SPARQL queries that are executed on the KB?
  * Does the same retriever work for both text and linearized KB triples?
* What is sparse retriever for text and KB categories in sec 4.2?
* Apart from the model details, the result sections are also very unclear. For example,
  * It is very unclear to me why ChatGPT which does not use any knowledge is the most competitive model in Table 3? I found an explanation of this missing in the paper.
  * Why does the text + kb + sparql model perform a little worse than the text + kb model? I found an explanation of this missing in the paper
  * Overall, it is unclear if the proposed model in the paper works better than ChatGPT even though the underlying LLM of the proposed model is also ChatGPT

**Questions:**

I have listed several questions in the Weakness section which needs to be addressed to make the paper better.

---

> ### Author Response · Authors · 2023-11-23
> **Response to Reviewer 7HFZ**
>
> We appreciate the insightful feedback provided on our submission. Your constructive comments have significantly contributed to refining our paper. Below, we address each point raised and explain the enhancements made in the updated version.
>
> ### 1. Dataset Construction
>
> **Weakness 1:**
> Even though I think the decision to include KB triples where the subject/object entity is not present in Wikidata to ensure equal reasoning importance to both KB and text is reasonable, but even though an entity is not captured in Wikidata, that doesn't necessarily mean that fact is not present in web-text. Since LLMs are trained on web-text, what is the guarantee that an LLM would require KB fact to answer the question? It is possible that the knowledge corresponding to the KB triplet is present in the LLM because it picked that up from a text snippet (not in Wikipedia) during pre-training.
>
> **Response:**
> Your observation rightly points out the potential gap in Wikidata where certain subject/object entities are not present, indicating that these entities may be part of what is often termed 'long-tail knowledge.' While it is true that such information might exist in the broader web text that LLMs are trained on, recent research [1] has shown that LLMs can struggle with recalling long-tail knowledge accurately. This is a significant insight, as it underlines a limitation in the ability of LLMs to retrieve less common or rarely mentioned information.
>
> Given this limitation, relying solely on an LLM's training on web text may not always be sufficient, especially for entities that fall into this long-tail category. This is where the integration of a knowledge base (KB) becomes essential. By retrieving information directly from a KB, we can supplement the potential gaps in an LLM's knowledge, especially for those less represented or niche topics.
>
> Therefore, in the context of answering questions from our curated dataset, incorporating KB retrieval becomes a necessary step. This approach not only enhances the accuracy of the information provided but also ensures that we are tapping into a wider spectrum of knowledge, covering areas that might be underrepresented in the LLM's training data.
>
> _Reference:_
> [1] _When Not to Trust Language Models: Investigating Effectiveness of Parametric and Non-Parametric Memories_ (Mallen et al., ACL 2023)
>
> ### 2. Clarity and Missing Details in Model Description
>
> **Weaknesses 2.a:** What are the instructions for query decomposition given to LLM?
>
> **Response:**
> In the revised paper (refer to Section 3), we have elaborated on the instructions for query decomposition provided to the LLM, ensuring a more comprehensive understanding of the process.
> We provide Figure 4 from Appendix A.8 as an illustration of DETLLM to instruct LLMs for addressing multi-source multi-hop questions. We also provide the exact prompt in Appendix A.3 that is used to prompt LLM in the proposed method.
>
> **Weaknesses 2.b:** How does the retriever work?
>
> **Response:**
> We include more details in Section 3.2 in the updated pdf. The retriever is used to retrieve relevant information from both knowledge sources. For unstructured knowledge, we use a dense retriever Contriever (Izacard et al., 2022) to retrieve relevant passages, while for structured knowledge, we retrieve relevant information from both textual and structured formats. For structured knowledge, we use a sparse text retriever BM25 (Robertson et al., 2009) to retrieve from linearized triplet.
>
> **Weaknesses 2.c:** Similarly, what is the model that generates SPARQL queries that are executed on the KB?
>
> **Response:**
> We use ChatGPT to translate a decomposed single-hop question to SPARQL. The prompt is shown in Appendix A.
>
> **Weaknesses 2.d:** Does the same retriever work for both text and linearized KB triples?
>
> **Response:**
> No, the same retriever is not equally effective for both text and linearized KB triples. As detailed in our ablation study in Section 4.3 and Table 4, we found that distinct retrievers are more suitable for different types of knowledge. The unsupervised dense retriever, like Contriever, which is trained on natural language corpora, is well-suited for unstructured text due to its adaptability in processing natural language. However, it falls short in handling structured knowledge like linearized KB triples, where natural language formatting is absent. On the other hand, the sparse retriever BM25 excels with structured knowledge, leveraging its keyword-based search mechanism. Therefore, we advocate for a tailored approach in retriever tool selection, depending on the nature of the knowledge source.
>
> ### 3. Q: What is sparse retriever for text and KB categories in sec 4.2?
>
> **Response:**
> The sparse retriever used is BM25[2].
>
> _Reference:_
> [2] _Stephen Robertson, Hugo Zaragoza, et al. 2009. The probabilistic relevance framework: Bm25 and beyond._ Foundations and Trends® in Information Retrieval, 3(4):333–389

---

> > ### Author Response · Authors · 2023-11-23
> > **continue**
> >
> > ### 4. Significance and Effectiveness of the Proposed Model
> >
> > **Weaknesses 4.a:**
> > It is very unclear to me why ChatGPT, which does not use any knowledge, is the most competitive model in Table 3? I found an explanation of this missing in the paper.
> >
> > **Response:**
> > In the updated Section 4.2 of our paper, we've provided an in-depth analysis comparing our DETLLM model with ChatGPT and other models, ensuring each has access to both KB and Text resources. This comparison, detailed in Table 2, demonstrates that DETLLM outperforms the others, including ChatGPT.
> >
> > Further, we delve into ChatGPT's performance in a closed-book setting in Section 4.3, 'Comparing with Closed-book LLM', with results presented in Table 6. While ChatGPT shows competitive performance, DETLLM displays notable improvements in specific scenarios. For instance, only about half of the examples correctly answered by DETLLM overlap with those answered by closed-book ChatGPT. This indicates that DETLLM offers a distinct advantage, handling a broader range of queries effectively. Overall, combining the strengths of both models covers approximately 45.4% of the entire dataset, underscoring the complementary nature of their capabilities.
> >
> > **Weaknesses 4.b:**
> > Why does the text + kb + sparql model perform a little worse than the text + kb model?
> >
> > **Response:**
> > We add an ablation study in Section 4.3. The results are shown in Table 3. We present the results of an ablation study involving three key factors: a) the integration of heterogeneous knowledge sources, b) the choice between dense and sparse retrievers, and c) the incorporation of SPARQL. Our findings indicate that optimal performance is achieved when handling heterogeneous knowledge sources separately, combined with careful retriever tool selection. The unsupervised dense retriever (i.e., Contriever), trained on natural language corpus, demonstrates adaptability to unstructured knowledge but loses its advantage when dealing with linearized structured knowledge due to the absence of natural language formatting. Conversely, the sparse retriever BM25 performs better on structured knowledge, relying on keyword-based search methodologies. Furthermore, the SPARQL tool consistently outperforms its counterparts in all settings, showcasing improvements regardless of the integration of knowledge sources and the choice of retriever.